# Optimal Neural Network Approximation of Wasserstein Gradient Direction via Convex Optimization

## Abstract

The computation of Wasserstein gradient direction is essential for posterior sampling problems and scientific computing. The approximation of the Wasserstein gradient with finite samples requires solving a variational problem. We study the variational problem in the family of two-layer networks with squared-ReLU activations, towards which we derive a semi-definite programming (SDP) relaxation. This SDP can be viewed as an approximation of the Wasserstein gradient in a broader function family including two-layer networks. By solving the convex SDP, we obtain the optimal approximation of the Wasserstein gradient direction in this class of functions. Numerical experiments including PDE-constrained Bayesian inference and parameter estimation in COVID-19 modeling demonstrate the effectiveness of the proposed method.

## 1 Introduction

Bayesian inference plays an essential role in learning model parameters from the observational data with applications in inverse problems, scientific computing, information science, and machine learning (Stuart, 2010). The central problem in Bayesian inference is to draw samples from a posterior distribution, which characterizes the parameter distribution given data and a prior distribution.

The Wasserstein gradient flow (Otto, 2001; Ambrosio et al., 2005; Junge et al., 2017) has shown to be effective in drawing samples from a posterior distribution, which attracts increasing attention in recent years. For instance, the Wasserstein gradient flow of Kullback-Leibler (KL) divergence connects to the overdampled Langevin dynamics. The time-discretization of the overdamped Langevin dynamics renders the classical Langevin Monte Carlo Markov Chain (MCMC) algorithm. In this sense, the computation of Wasserstein gradient flow yields a different viewpoint for sampling algorithms. In particular, the Wasserstein gradient direction also provides a deterministic update of the particle system (Carrillo et al., 2021b). Based on the approximation or generalization of the Wasserstein gradient direction, many efficient sampling algorithms have been developed, including Wasserstein gradient descent (WGD) with kernel density estimation (KDE) (Liu et al., 2019), Stein variational gradient descent (SVGD) (Liu & Wang, 2016), and neural variational gradient descent (di Langosco et al., 2021), etc.

Meanwhile, neural networks exhibit tremendous optimization and generalization performance in learning complicated functions from data. They also have wide applications in Bayesian inverse problems (Rezende & Mohamed, 2015; Onken et al., 2020; Kruse et al., 2019; Lan et al., 2021). According to the universal approximation theorem of neural networks (Hornik et al., 1989; Lu et al., 2017), any arbitrarily complicated functions can be learned by a two-layer neural network with

34 non-linear activations and a sufficient number of neurons. Functions represented by neural networks
35 naturally provide an approximation towards the Wasserstein gradient direction.

36 However, due to the nonlinear and nonconvex structure of neural networks, optimization algorithms
37 including stochastic gradient descent may not find the global optima of the training problem. Recently,
38 based on a line of works (Pilanci & Ergen, 2020; Sahiner et al., 2020; Bartan & Pilanci, 2021), the
39 regularized training problem of two-layer neural networks with ReLU/polynomial activation can
40 be formulated as a convex program. The optimal solution of the convex program renders a global
41 optimum of the nonconvex training problem.

42 In this paper, we study a variational problem, whose optimal solution corresponds to the Wasserstein
43 gradient direction. Focusing on the family of two-layer neural networks with squared ReLU activation,
44 we formulate the regularized variational problem in terms of samples. Directly training the neural
45 network to minimize the loss may get the neural network stuck at local minima or saddle points and
46 it often leads to biased sample distribution from the posterior. Instead, we analyze the convex dual
47 problem of the training problem and study its semi-definite program (SDP) relaxation by analyzing
48 the geometry of dual constraints. The resulting SDP is practically solvable and it can be efficiently
49 optimized by convex optimization solvers such as CVXPY (Diamond & Boyd, 2016). We then derive
50 the corresponding relaxed bidual problem (dual of the relaxed dual problem). Thus, the optimal
51 solution to the dual problem yields an optimal approximation of the Wasserstein gradient direction
52 in a broader function family. We also present a practical implementation and analyze the choice of
53 the regularization parameter. Numerical results including PDE-constrained inference problems and
54 Covid-19 parameter estimation problems illustrate the effectiveness of our proposed method.

## 1.1 Related works

56 The time and spatial discretizations of Wasserstein gradient flows are extensively studied in literature
57 (Jordan et al., 1998; Junge et al., 2017; Carrillo et al., 2021a,b; Bonet et al., 2021; Liutkus et al., 2019;
58 Frogner & Poggio, 2020). Recently, neural networks have been applied in solving or approximating
59 Wasserstein gradient flows (Mokrov et al., 2021; Lin et al., 2021b,a; Alvarez-Melis et al., 2021;
60 Bunne et al., 2021; Hwang et al., 2021; Fan et al., 2021). For sampling algorithms, di Langosco
61 et al. (2021) learns the transportation function by solving an unregularized variational problem in the
62 family of vector-output deep neural networks. Compared to these studies, we focus on a convex SDP
63 relaxation of the varitional problem induced by the Wasserstein gradient direction. Meanwhile, Feng
64 et al. (2021) form the Wasserstein gradient direction as the minimizer the Bregman score and they
65 apply deep neural networks to solve the induced variational problem.

## 2 Background

67 In this section, we briefly review the Wasserstein gradient descent and present its variational for-
68 mulation. In particular, we focus on the Wasserstein gradient descent direction of KL divergence
69 functional. Later on, we design a neural network convex optimization problems to approximate the
70 Wasserstein gradient in samples.

### 2.1 Wasserstein gradient descent

72 Consider an optimization problem in the probability space:

$$\inf_{\rho \in \mathcal{P}} \mathrm{D_{KL}}(\rho \| \pi) = \int \rho(x)(\log \rho(x) - \log \pi(x))dx, \tag{1}$$

73 Here the integral is taken over $\mathbb{R}^d$ and the objective functional $\mathrm{D_{KL}}(\rho \| \pi)$ is the KL divergence from
74 $\rho$ to $\pi$. The variable is the density function $\rho$ in the space $\mathcal{P} = \{\rho \in C^\infty(\mathbb{R}^d) | \int \rho dx = 1, \ \rho > 0\}$.
75 The function $\pi \in C^\infty(\mathbb{R}^d)$ is a known probability density function of the posterior distribution. By
76 solving the optimization problem (1), we can generate samples from the posterior distribution.

A known fact (Villani, 2003, Chapter 8.3.1) is that the Wasserstein gradient descent flow for the optimization problem (1) satisfies

$$
\begin{aligned}
\partial_t \rho_t =& \nabla \cdot \left( \rho_t \nabla \frac{\delta}{\delta \rho_t} \mathrm{D_{KL}}(\rho_t \| \pi) \right) \\
=& \nabla \cdot (\rho_t (\nabla \log \rho_t - \nabla \log \pi)) \\
=& \Delta \rho_t - \nabla \cdot (\rho_t \nabla \log \pi),
\end{aligned}
$$

where $\rho_t(x) = \rho(x, t)$ and $\frac{\delta}{\delta \rho_t}$ is the $L^2$ first variation operator w.r.t. $\rho_t$. In the above third equality, a fact $\rho_t \nabla \log \rho_t = \nabla \rho_t$ is used. Here $\nabla \cdot F$ denotes the divergence of a vector valued function $F : \mathbb{R}^d \to \mathbb{R}^d$ and $\Delta$ is the Laplace operator. This equation is also known as the gradient drift Fokker-Planck equation. It corresponds to the following updates in terms of samples:

$$
dx_t = -(\nabla \log \rho_t(x_t) - \nabla \log \pi(x_t))dt, \tag{2}
$$

where $x_t$ follows the distribution of $\rho_t$. Clearly, when $\rho_t = \pi$, the above dynamics reach the equilibrium, which implies that the samples $x_t$ are generated by the posterior distribution.

To solve the Wasserstein gradient flow (2), we consider a forward Eulerian discretization in time. In the $l$-th iteration, suppose that $\{x_l^n\}$ are samples drawn from $\rho_l$. The update rule of Wasserstein gradient descent (WGD) on the particle system $\{x_l^n\}$ follows

$$
x_{l+1}^n = x_l^n - \alpha_l \nabla \Phi_l(x_l^n), \tag{3}
$$

where $\Phi_l : \mathbb{R}^d \to \mathbb{R}$ is a function which approximates $\log \rho_l - \log \pi$ and $\alpha_l > 0$ is the step size.

## 2.2 Variational formulation of WGD

Given the particles $\{x_n\}_{n=1}^N$, we design the following variational problem to choose a suitable function $\Phi$ approximating the function $\log \rho - \log \pi$. Consider

$$
\inf_{\Phi \in C^1(\mathbb{R}^d)} \frac{1}{2} \int \|\nabla \Phi(x - (\nabla \log \rho(x) - \nabla \log \pi(x))\|_2^2 \rho(x)dx. \tag{4}
$$

The objective functional evaluates the least-square discrepancy between $\nabla \log \rho - \nabla \log \pi$ and $\nabla \Phi$ weighted by the density $\rho$. The optimal solution follows $\Phi = \log \rho - \log \pi$, up to a constant shift. Let $\mathcal{H} \subseteq C^1(\mathbb{R}^d)$ be a finite dimensional function space. The following proposition gives a formulation of (4) in $\mathcal{H}$.

**Proposition 1** *Let $\mathcal{H} \subseteq C^1(\mathbb{R}^d)$ be a function space. The variational problem (4) in the domain $\mathcal{H}$ is equivalent to*

$$
\inf_{\Phi \in \mathcal{H}} \frac{1}{2} \int \|\nabla \Phi(x)\|_2^2 \rho dx + \int \Delta \Phi(x) \rho(x)dx + \int \langle \nabla \log \pi(x), \nabla \Phi(x) \rangle \rho(x)dx. \tag{5}
$$

**Remark 1** A similar variational problem has been studied in (di Langosco et al., 2021). If we replace $\nabla \Phi$ for $\Phi \in \mathcal{H}$ by a vector field $\Psi$ in certain function family, then, the quantity in (5) is the negative regularized Stein discrepancy defined in (di Langosco et al., 2021) between $\rho$ and $\pi$ based on $\Psi$. This problem is also similar to the varitional problem for the score matching estimator in (Hyvärinen & Dayan, 2005) by parameterizing $\Phi$ in a given probabilistic model. In comparison, our method can be viewed as a special case of score matching by using a two-layer neural network model.

Therefore, by replacing the density $\rho$ by finite samples $\{x_n\}_{n=1}^N \sim \rho$, the problem (5) in terms of finite samples forms

$$
\inf_{\Phi \in \mathcal{H}} \frac{1}{N} \sum_{n=1}^N \left( \frac{1}{2} \|\nabla \Phi(x_n)\|_2^2 + \Delta \Phi(x_n) \right) + \frac{1}{N} \sum_{n=1}^N \langle \nabla \log \pi(x_n), \nabla \Phi(x_n) \rangle. \tag{6}
$$

## 3 Optimal neural network approximation of Wasserstein gradient

In this section, we focus on functional space $\mathcal{H}$ of functions represented by two-layer neural networks. We derive the primal and dual problem of the regularized Wasserstein variational problems. By

analyzing the dual constraints, a convex SDP relaxation of the dual problem is obtained. We also present a practical implementation estimation of $\nabla \log \rho - \nabla \log \pi$ and discuss the choice of the regularization parameter.

Let $\psi$ be an activation function. Consider the case where $\mathcal{H}$ is a class of two-layer neural network with the activation function $\psi(x)$:

$$\mathcal{H} = \left\{ \Phi_{\boldsymbol{\theta}} \in C^1(\mathbb{R}^d) | \Phi_{\boldsymbol{\theta}}(x) = \alpha^T \psi(W^T x) \right\}, \tag{7}$$

where $\boldsymbol{\theta} = (W, \alpha)$ is the parameter in the neural network with $W \in \mathbb{R}^{d \times m}$ and $\alpha \in \mathbb{R}^m$.

**Remark 2** We can extend this model to handle the bias term by add an entry of 1 in $x_1, \ldots, x_n$.

For two-layer neural networks, we can compute the gradient and Laplacian of $\Phi \in \mathcal{H}$ as follows:

$$\nabla \Phi_{\boldsymbol{\theta}}(x) = \sum_{i=1}^{m} \alpha_i w_i \psi'(w_i^T x) = W(\psi'(W^T x) \circ \alpha), \tag{8}$$

$$\Delta \Phi_{\boldsymbol{\theta}}(x) = \sum_{i=1}^{m} \alpha_i \|w_i\|_2^2 \psi''(w_i^T x). \tag{9}$$

Here $\circ$ represents the element-wise multiplication. By adding a regularization term to the variational problem (6), we obtain

$$\min_{\boldsymbol{\theta}} \frac{1}{2N} \sum_{n=1}^{N} \left\| \sum_{i=1}^{m} \alpha_i w_i \psi'(w_i^T x_n) \right\|_2^2 + \frac{1}{N} \sum_{n=1}^{N} \left\langle \sum_{i=1}^{m} \alpha_i w_i \psi'(w_i^T x_n), \nabla \log \pi(x_n) \right\rangle$$
$$+ \frac{1}{N} \sum_{n=1}^{N} \sum_{i=1}^{m} \alpha_i \|w_i\|_2^2 \psi''(w_i^T x_n) + \frac{\beta}{2} R(\boldsymbol{\theta}), \tag{10}$$

where $\beta > 0$ is the regularization parameter. We focus on the squared ReLU activation $\psi(z) = (z)_+^2 = (\max\{z, 0\})^2$. Note that a non-vanishing second derivative is required for the Laplacian term in (9), which makes the ReLU activation inadequate. For this activation function, we consider the regularization function $R(\boldsymbol{\theta}) = \sum_{i=1}^{m} (\|w_i\|_2^3 + |\alpha_i|^3)$.

**Remark 3** We note that $\nabla \Phi_{\boldsymbol{\theta}}(x)$ and $\Delta \Phi_{\boldsymbol{\theta}}(x)$ are all piece-wise degree-3 polynomials of the parameters $\boldsymbol{\theta}$. Hence, we consider a specific cubic regularization term above, analogous to (Bartan & Pilanci, 2021). By choosing this regularization term, we can derive a simplified convex dual problem.

By rescaling the first and second-layer parameters, the regularized variational problem (10) can be formulated as follows.

**Proposition 2 (Primal problem)** *The regularized variational problem* (10) *is equivalent to*

$$\min_{W, \alpha} \frac{1}{2} \sum_{n=1}^{N} \left\| \sum_{i=1}^{m} \alpha_i w_i \psi'(w_i^T x_n) \right\|^2 + \sum_{n=1}^{N} \sum_{i=1}^{m} \alpha_i \|w_i\|_2^2 \psi''(w_i^T x_n)$$
$$+ \sum_{n=1}^{N} \left\langle \sum_{i=1}^{m} \alpha_i w_i \psi'(w_i^T x_n), \nabla \log \pi(x_n) \right\rangle + \tilde{\beta} \|\alpha\|_1, \tag{11}$$
$$s.t. \ \|w_i\|_2 \le 1, i \in [m],$$

*where* $\tilde{\beta} = 3 \cdot 2^{-5/3} N \beta$.

For simplicity, we write $Y = \begin{bmatrix} \nabla \log \pi(x_1)^T \\ \vdots \\ \nabla \log \pi(x_N)^T \end{bmatrix} \in \mathbb{R}^{N \times d}$. We introduce the slack variable $z_n = \sum_{i=1}^{m} \alpha_i w_i \psi'(x_n^T w_i)$ for $n \in [N]$ and denote $Z = [z_1 \ \ldots \ z_N]^T \in \mathbb{R}^{N \times d}$. Then, we can simplify the problem (11) to

$$\min_{W, \alpha, Z} \frac{1}{2} \|Z\|_F^2 + \sum_{n=1}^{N} \sum_{i=1}^{m} \alpha_i \|w_i\|_2^2 \psi''(w_i^T x_n) + \text{tr}(Y^T Z) + \tilde{\beta} \|\alpha\|_1, \tag{12}$$
$$s.t. \ z_n = \sum_{i=1}^{m} \alpha_i w_i \psi'(x_n^T w_i), n \in [N], \|w_i\|_2 \le 1, i \in [m].$$

Based on the above reformulation, we can derive the dual problem of (12) as follows.

**Proposition 3 (Dual problem)** *The dual problem of the regularized variational problem* (12) *is*

$$\max_{\Lambda \in \mathbb{R}^{N \times d}} -\frac{1}{2}\|\Lambda + Y\|_F^2, \; s.t. \; \max_{w:\|w\|_2 \leq 1} \left| \sum_{n=1}^N \|w\|_2^2 \psi''(x_n^T w) - \lambda_n^T w \psi'(x_n^T w) \right| \leq \tilde{\beta}, \qquad (13)$$

*which provides a lower-bound on* (12)*.*

### 3.1 Analysis of dual constraints and the relaxed dual problem

Now, we analyze the constraint

$$\max_{w:\|w\|_2 \leq 1} \left| \sum_{n=1}^N \|w\|_2^2 \psi''(w^T x_n) - \lambda_n^T w \psi'(x_n^T w) \right| \leq \tilde{\beta}$$

in the dual problem. We note that this constraint is closely related to the regularization parameter, which we will discuss later. For simplicity, we take $\psi''(0) = 0$ as the subgradient of $\psi'(z)$ at $z = 0$, i.e., taking the left derivative of $\psi'(z)$ at $z = 0$. Let $X = [x_1, \ldots, x_N]^T \in \mathbb{R}^{N \times d}$. Denote the set of all possible hyper-plane arrangements corresponding to the rows of $X$ as

$$\mathcal{S} = \{D = \mathbf{diag}(\mathbb{I}(Xw \geq 0)) | w \in \mathbb{R}^d, w \neq 0\}. \qquad (14)$$

Here $\mathbb{I}(s) = 1$ if the statement $s$ is correct and $\mathbb{I}(s) = 0$ otherwise. Let $p = |\mathcal{S}|$ be the cardinality of $\mathcal{S}$, and write $\mathcal{S} = \{D_1, \ldots, D_p\}$. According to (Cover, 1965), we have the upper bound $p \leq 2r \left( \frac{e(N-1)}{r} \right)^r$, where $r = \text{rank}(X)$.

Based on the analysis of the dual constraints, we can derive a convex SDP as a relaxed dual problem. It gives a lower bound for the optimal value of the dual problem (13).

**Proposition 4 (Relaxed Dual problem)** *Consider the following SDP:*

$$\max \; -\frac{1}{2}\|\Lambda + Y\|_F^2,$$

$$s.t. \; \tilde{A}_j(\Lambda) + \tilde{B}_j + \sum_{n=0}^N r_n^{(j,-)} H_n^{(j)} + \tilde{\beta} e_{d+1} e_{d+1}^T \succeq 0,$$

$$-\tilde{A}_j(\Lambda) - \tilde{B}_j + \sum_{n=0}^N r_n^{(j,+)} H_n^{(j)} + \tilde{\beta} e_{d+1} e_{d+1}^T \succeq 0, \qquad (15)$$

$$r^{(j,-)} \geq 0, r^{(j,+)} \geq 0, j \in [p].$$

*The variables are* $\Lambda \in \mathbb{R}^{N \times d}$ *and* $r^{(j,-)}, r^{(j,+)} \in \mathbb{R}^{n+1}$ *for* $j \in [p]$. *For* $j \in [p]$, *we denote* $A_j(\Lambda) = -\Lambda^T D_j X - X^T D_j \Lambda$, $B_j = 2\,\text{tr}(D_j) I_d$, $\tilde{A}_j(\Lambda) = \begin{bmatrix} A_j(\Lambda) & 0 \\ 0 & 0 \end{bmatrix}$, $\tilde{B}_j = \begin{bmatrix} B_j & 0 \\ 0 & 0 \end{bmatrix}$, $H_0^{(j)} = \begin{bmatrix} I_d & 0 \\ 0 & -1 \end{bmatrix}$ *and* $H_n^{(j)} = \begin{bmatrix} 0 & (1-2(D_j)_{nn})x_n \\ (1-2(D_j)_{nn})x_n^T & 0 \end{bmatrix}$, $n \in [N]$ *The vector* $e_{d+1} \in \mathbb{R}^{d+1}$ *satisfies that* $(e_{d+1})_i = 0$ *for* $i \in [d]$ *and* $(e_{d+1})_{d+1} = 1$.

*The optimal value of* (15) *gives a lower bound on the dual problem* (13), *and hence on the primal problem* (12).

In the following proposition, we derive the relaxed bi-dual problem. It can be viewed as a convex relaxation of the primal problem (12).

**Proposition 5 (Relaxed bi-dual problem)** *The dual of the relaxed dual problem* (15) *is as follows*

$$\min \frac{1}{2}\|Z + Y\|_F^2 - \frac{1}{2}\|Y\|_F^2 + \sum_{j=1}^p \text{tr}(\tilde{B}_j(S^{(j,+)} - S^{(j,-)})) + \tilde{\beta} \sum_{j=1}^p \text{tr}\left((S^{(j,+)} + S^{(j,-)})e_{d+1}e_{d+1}^T\right),$$

$$s.t. \; Z = \sum_{j=1}^p \tilde{A}_j^*(S^{(j,-)} - S^{(j,+)}), \text{tr}(S^{(j,-)} H_n^{(j)}) \leq 0, \text{tr}(S^{(j,+)} H_n^{(j)}) \leq 0, n = 0, \ldots, N, j \in [p],$$

$$(16)$$

*in variables $Z \in \mathbb{R}^{N \times d}$, $S^{(j,+)}, S^{(j,-)} \in \mathbb{S}_+^{d+1}$ for $j \in [p]$. Here $A_j^*$ is the adjoint operator of the linear operator $A_j$.*

As (15) is a convex problem and the Slater's condition is satisfied, the optimal values of (15) and (16) are same. We can show that any feasible solutions of the primal problem (11) can be mapped to feasible solutions of (16).

**Theorem 1** *Suppose that $(Z, W, \alpha)$ is feasible to the primal problem (12). Then, there exist matrices $\{S^{(j,+)}, S^{(j,-)}\}_{j=1}^p$ constructed from $(W, \alpha)$ such that $(Z, \{S^{(j,+)}, S^{(j,-)}\}_{j=1}^p)$ is feasible to the relaxed bi-dual problem (16). Moreover, the objective value of the relaxed bi-dual problem (16) at $(Z, \{S^{(j,+)}, S^{(j,-)}\}_{j=1}^p)$ is the same as objective value of the primal problem (12) at $(Z, W, \alpha)$.*

Let $J(Z, \{S^{(j,+)}, S^{(j,-)}\}_{j=1}^p)$ denote the objective value of the relaxed bi-dual problem (16) at a feasible solution $(Z, \{S^{(j,+)}, S^{(j,-)}\}_{j=1}^p)$. Let $(Z^*, W^*, \alpha^*)$ denote a globally optimal solution of the primal problem (12). By Theorem 1, there exist matrices $\{S^{(j,+)}, S^{(j,-)}\}_{j=1}^p$ such that $(Z^*, \{S^{(j,+)}, S^{(j,-)}\}_{j=1}^p)$ is a feasible solution of the relaxed bi-dual problem (16) and $J(Z^*, \{S^{(j,+)}, S^{(j,-)}\}_{j=1}^p)$ is the same as the objective value of (12) at its global minimum $(Z^*, W^*, \alpha^*)$. On the other hand, let $(\tilde{Z}^*, \{\tilde{S}^{(j,+)}, \tilde{S}^{(j,-)}\}_{j=1}^p)$ denote an optimal solution of the relaxed bi-dual problem (16). From the optimality of $(\tilde{Z}^*, \{\tilde{S}^{(j,+)}, \tilde{S}^{(j,-)}\}_{j=1}^p)$, we have

$$J(\tilde{Z}^*, \{\tilde{S}^{(j,+)}, \tilde{S}^{(j,-)}\}_{j=1}^p) \le J(Z^*, \{S^{(j,+)}, S^{(j,-)}\}_{j=1}^p). \tag{17}$$

Note that at $(Z^*, W^*, \alpha^*)$ we obtain the optimal approximation of $\nabla \log \rho - \nabla \log \pi$ at $x_1, \dots, x_N$ in the family of two-layer squared-ReLU networks (7). Smaller or equal objective value of the relaxed bi-dual problem (16) can be achieved at $(\tilde{Z}^*, \{\tilde{S}^{(j,+)}, \tilde{S}^{(j,-)}\}_{j=1}^p)$ than at $(Z^*, \{S^{(j,+)}, S^{(j,-)}\}_{j=1}^p)$. Therefore, we can view $\tilde{Z}^*$ gives an optimal approximation of $\nabla \log \rho - \nabla \log \pi$ evaluated on $x_1, \dots, x_N$ in a broader function family including the two-layer squared ReLU neural networks.

From the derivation of the relaxed bi-dual problem, we have the relation $\tilde{Z}^* = -\Lambda^* - Y$, where $(\Lambda^*, \{r^{(j,+)}, r^{(j,-)})$ is optimal to the relaxed dual problem (15) and $(\tilde{Z}^*, \{\tilde{S}^{(j,+)}, \tilde{S}^{(j,-)}\}_{j=1}^p)$ is optimal to the relaxed bi-dual problem (16). Therefore, by solving $\Lambda^*$ from the relaxed dual problem (15), we can use $-\Lambda^* - Y$ as the approximation of $\nabla \log \rho - \nabla \log \pi$ evaluated on $x_1, \dots, x_N$.

**Remark 4** We note that solving the proposed convex optimization problem 15 renders the approximation of the Wasserstein gradient direction. Compared to the two-layer ReLU networks, it induces a broader class of functions represented by $\{S^{(j,+)}, S^{(j,-)}\}_{j=1}^p$. This contains more variables than the neural network function.

## 3.2 Practical implementation

Although the number $p$ of all possible hyper-plane arrangements is upper bounded by $2r((N-1)e/r)^r$ with $r = \text{rank}(X)$, it is computationally costly to enumerate all possible $p$ matrices $D_1, \dots, D_p$ to represent the constraints in the relaxed dual problem (4). In practice, we first randomly sample $M$ i.i.d. random vectors $u_1, \dots, u_M \sim \mathcal{N}(0, I_d)$ and generate a subset $\hat{S}$ of $S$ as follows:

$$\hat{S} = \{\textbf{diag}(\mathbb{I}(Xu_j \ge 0)|j \in [M]\}. \tag{18}$$

Then, we optimize the randomly sub-sampled version of the relaxed dual problem based on the subset $\hat{S}$ and obtain the solution $\Lambda$. We then use $-\Lambda - Y$ as the direction to update the particle system $X$.

If the regularization parameter is too large, then we will have $-\Lambda - Y = 0$, which makes the particle system unchanged. Therefore, to ensure that $\tilde{\beta}$ is not too large, we decay $\tilde{\beta}$ by a factor $\gamma_1 \in (0, 1)$. This also appears in (Ergen et al., 2021). On the other hand, if $\tilde{\beta}$ is too small resulting the relaxed dual problem (4) infeasible, we increase $\tilde{\beta}$ by multiplying $\gamma_2^{-1}$, where $\gamma_2 \in (0, 1)$. Detailed explanation of the adjustment of the regularization parameter can be found in Appendix C. The overall algorithm is summarized in Algorithm 1.

We note that the randomly subsampled version of the relaxed dual problem (15) involves $2N\hat{p}$ inequality constraints and $2\hat{p}$ linear matrix inequality constraints with size $(d + 1) \times (d + 1)$. Applying the standard interior point method (Boyd et al., 2004) leads to the computational time up to

$$O((\max\{N, d^2\}\hat{p})^6).$$

---
**Algorithm 1** Convex neural Wasserstein descent
---
**Require:** initial positions $\{x_0^n\}_{n=1}^N$, step size $\alpha_l$, initial regularization parameter $\tilde{\beta}_0, \gamma_1, \gamma_2 \in (0,1)$.

---

  1: **while** not converge **do**
  2:      Form $X_l$ and $Y_l$ based on $\{x_l^n\}_{n=1}^N$ and $\{\nabla \log \pi(x_l^n)\}_{n=1}^N$.
  3:      Solve $\Lambda_l$ from the relaxed dual problem (15) with $\tilde{\beta} = \tilde{\beta}_l$.
  4:      **if** the relaxed dual problem with $\tilde{\beta} = \tilde{\beta}_l$ is infeasible **then**
  5:          Set $X_{l+1} = X_l$ for $n \in [N]$ and set $\tilde{\beta}_{l+1} = \gamma_2^{-1}\tilde{\beta}_l$.
  6:      **else**
  7:          Update $X_{l+1} = X_l + \alpha_l(\Lambda_l + Y_l)$ for $n \in [N]$ and set $\tilde{\beta}_{l+1} = \gamma_1\tilde{\beta}_l$.
  8:      **end if**
  9: **end while**

---

For high-dimensional problems, i.e., $d$ is large, the computational cost of solving (15) can be large. In this case, we apply the dimension-reduction techniques (Zahm et al., 2018; Chen & Ghattas, 2020; Wang et al., 2021) to reduce the parameter dimension $d$ to a data-informed intrinsic dimension $\hat{d}$, which is often very low, i.e., $\hat{d} \ll d$.

## 4 Numerical experiments

In this section, we present numerical results to compare WGD approximated by neural networks (WGD-NN) and WGD approximated using convex optimization formulation of neural networks (WGD-cvxNN). The performance of the two methods is assessed by the sample goodness-of-fit of the posterior. For WGD-NN, in each iteration, it updates the particle system using (3) with a function $\Phi$ represented by a two-layer squared ReLU neural network. The parameters of the neural network is obtained by directly solving the nonconvex optimization problem (10). We note that it takes longer time by WGD-cvxNN (compared to WGD-NN) to solve the convex optimization problem. However, this optimization time is often dominated by the time in likelihood evaluation if the model is expensive to solve. Moreover, the induced SDPs have specific structures of many similar constraints, whose solution can be accelerated by designing a specialized convex optimization solver. This is left for future work.

### 4.1 A toy example

We test the performance of WGD on a bimodal 2-dimensional double-banana posterior distribution introduced in (Detommaso et al., 2018). We first generate 300 posterior samples by a Stein variational Newton (SVN) method (Detommaso et al., 2018) as the reference, as shown in Figure 1. We evaluate the performance of WGD-NN and WGD-cvxNN by calculating the maximum mean discrepancy (MMD) between their samples in each iteration and the reference samples. In the comparison, we use $N = 50$ samples and run for 100 iterations with step sizes $\alpha_l = 10^{-3}$. For WGD-cvxNN, we set $\beta = 1$, $\gamma_1 = 0.95$ and $\gamma_2 = 0.95^{10}$. For WGD-NN, we use $m = 200$ neurons and optimize the regularized training problem (10) using all samples with the Adam optimizer (Kingma & Ba, 2014) with learning rate $10^{-3}$ for 200 sub-iterations. We also set the regularization parameter $\beta = 1$ and decrease it by a factor of 0.95 in each iteration. We find that this setup of parameters is more suitable.

The posterior density and the sample distributions by WGD-cvxNN and WGD-NN at the final step of 100 iterations are shown in Figure 1. It can be observed that WGD-cvxNN provides more representative samples than WGD-NN for the posterior density.

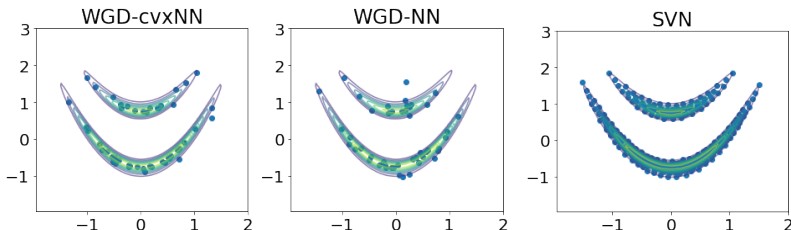

Figure 1: Posterior density and sample distributions by WGD-cvxNN and WGD-NN at the final step of 100 iterations, compared to the reference SVN samples (right).

In Figure 2, we plot the MMD of the samples by WGD-cvxNN and WGD-NN compared to the reference SVN samples at each iteration. We observe that the samples by WGD-cvxNN achieves much smaller MMD than those of WGD-NN compared to the reference SVN samples, which is consistent with the results shown in Figure 1. For WGD-cvxNN, it takes 572s in total, while for WGD-NN, it takes 16s in total. WGD-cvxNN takes much longer time than WGD-NN as WGD-cvxNN aims to solve for the global minimum of the relaxed convex dual problem.

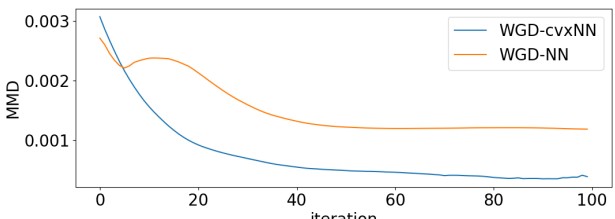

Figure 2: MMD of WGD-cvxNN and WGD-NN samples compared to the reference SVN samples.

## 4.2 PDE-constrained nonlinear Bayesian inference

In this experiment, we consider a nonlinear Bayesian inference problem constrained by the following partial differential equation (PDE) (Chen & Ghattas, 2020) with application to subsurface (Darcy) flow in a physical domain $D = (0, 1)^2$,

$$
\begin{aligned}
\mathbf{v} + e^x \nabla u &= 0 \quad \text{in } D, \\
\nabla \cdot \mathbf{v} &= h \quad \text{in } D,
\end{aligned} \tag{19}
$$

where $u$ is pressure, $\mathbf{v}$ is velocity, $h$ is force, $e^x$ is a random (permeability) field equipped with a Gaussian prior $x \sim \mathcal{N}(x_0, C)$ with covariance operator $C = (-\delta \Delta + \gamma I)^{-\alpha}$ where we set $\delta = 0.1, \gamma = 1, \alpha = 2$ and $x_0 = 0$. This problem is widely used in many areas, for instance, estimating permeability in groundwater flow, thermal conductivity in material science or electrical impedance in medical imaging, We impose Dirichlet boundary conditions $u = 1$ on the top boundary and $u = 0$ on the bottom boundary, and homogeneous Neumann boundary conditions on the left and right boundaries for $u$. We use a finite element method with piecewise linear elements for the discretization of the problem, resulting in 81 dimensions for the discrete parameter. The data is generated as pointwise observation of the pressure field at 49 points equidistantly distributed in $(0, 1)^2$, corrupted with additive $5\%$ Gaussian noise. We use a DILI-MCMC algorithm Cui et al. (2016) with 10000 effective samples to compute the sample mean and sample variance, which are used as the reference values to assess the goodness of the samples by pWGD-cvxNN and pWGD-NN.

We run pWGD-cvxNN and pWGD-NN with 64 samples for ten trials with step size $\alpha_l = 10^{-3}$, where we set $\beta = 10, \gamma_1 = 0.95$, and $\gamma_2 = 0.95^{10}$ for both methods. The RMSE of the sample mean and sample variance are shown in Figure 3 for the two methods at each of the iterations. We can observe that pWGD-cvxNN achieves smaller errors for both the sample mean and the sample variance compared to pWGD-NN at each iteration. Moreover, pWGD-cvxNN provides much smaller variation of the sample mean and sample variance for the ten trials compared to pWGD-NN.

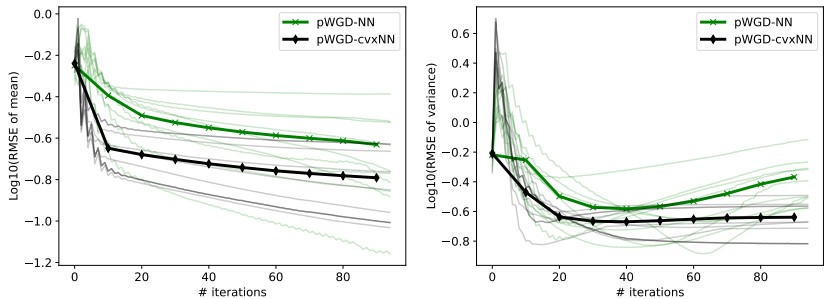

Figure 3: Ten trials and the RMSE of the sample mean (top) and sample variance (bottom) by pWGD-NN and pWGD-cvxNN at different iterations. Nonlinear inference problem.

### 4.3 Bayesian inference for COVID-19

In this experiment, we use Bayesian inference to learn the dynamics of the transmission and severity of COVID-19 from the recorded data for New York state, as studied in Chen & Ghattas (2020). We use the model, parameter, and data as in Chen & Ghattas (2020). More specifically, we use a compartmental model for the modeling of the transmission and outcome of COVID-19. We take the number of hospitalized cases as the observation data to infer a social distancing parameter, a time-dependent stochastic process that is equipped with a Tanh–Gaussian prior to model the transmission reduction effect of social distancing, which becomes 96 dimensions after discretization.

We run a projected Stein variational gradient descent (pSVGD) method Chen & Ghattas (2020) as the reference, and run pWGD-cvxNN and pWGD-NN using 64 samples for 100 iterations with step size $\alpha_l = 10^{-3}$, where we set $\beta = 10$, $\gamma_1 = 0.95$, and $\gamma_2 = 0.95^{10}$ for both methods as in the last example. From Figure 4 we can observe that pWGD-cvxNN produces more consistent results with pSVGD than pWGD-NN for both the sample mean and 90% credible interval, both in the inference of the social distancing parameter and in the prediction of the hospitalized cases.

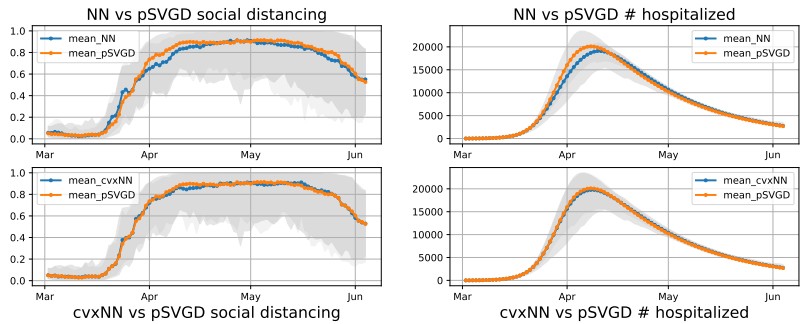

Figure 4: Comparison of pWGD-cvxNN and pWGD-NN to the reference by pSVGD for Bayesian inference of the social distancing parameter (left) from the data of the hospitalized cases (right) with sample mean and 90% credible interval.

## 5 Conclusion

In the context of variational Wasserstein gradient descent methods for Bayesian inference, we consider the approximation of the Wasserstein gradient direction by the gradient of functions in the family of two-layer neural networks. We propose a convex SDP relaxation of the dual of the variational primal problem, which can be solved efficiently using convex optimization methods instead of directly training the neural network as a nonconvex optimization problem. In particular, we established that the gradient obtained by the new formulation and convex optimization is at least as good as the optimal approximation of the Wasserstein gradient direction by functions in the family of two-layer neural networks, which is demonstrated by various numerical experiments. In future works, we expect to extend our convex neural network approximations to generalized Wasserstein flows.

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
