## A    Codes for numerical experiment

All codes for the numerical experiment can be found in `https://github.com/ai-submit/OptimalWasserstein`.

## B    Additional numerical experiment

### B.1    PDE-constrained linear Bayesian inference

In this experiment, we consider a linear Bayesian inference problem constrained by a partial differential equation (PDE) model for contaminant diffusion in environmental engineering in domain $D = (0, 1)$,

$$-\kappa \Delta u + \nu u = x \quad \text{in } D,$$

where $x$ is a contaminant source field parameter in domain $D$, $u$ is the contaminant concentration which we can observe at some locations, $\kappa$ and $\nu$ are diffusion and reaction coefficients. For simplicity, we set $\kappa, \nu = 1$, $u(0) = u(1) = 0$, and consider 15 pointwise observations of $u$ with $1\%$ noise, equidistantly distributed in $D$. We consider a Gaussian prior distribution $x \sim \mathcal{N}(0, C)$ with covariance given by a differential operator $C = (-\delta\Delta + \gamma I)^{-\alpha}$ with $\delta, \gamma, \alpha > 0$ representing the correlation length and variance, which is commonly used in geoscience. We set $\delta = 0.1, \gamma = 1, \alpha = 1$. In this linear setting, the posterior is Gaussian with the mean and covariance given analytically, which are used as reference to assess the sample goodness. We solve this forward model by a finite element method with piece-wise linear elements on a uniform mesh of size $2^k$, $k \geq 1$. We project this high-dimensional parameter to the data-informed low dimensions as in Wang et al. (2021) to alleviate the curse of dimensionality when applying WGD-cvxNN and WGD-NN, which we call pWGD-cvxNN and pWGD-NN, respectively. For $k = 4$ we have 17 dimensions for the discrete parameter and 4 dimensions after projection.

We run pWGD-cvxNN and pWGD-NN using 16 samples for 200 iterations with $\alpha_l = 10^{-3}$, $\beta = 5$, $\gamma_1 = 0.95$, and $\gamma_2 = 0.95^{10}$ for both methods. We use $m = 200$ neurons for pWGD-NN and train it by the Adam optimizer for 200 sub-iterations as in the first example. From Figure 5, we observe that pWGD-cvxNN achieves better root mean squared error (RMSE) than pWGD-NN for both the sample mean and the sample variance compared to the reference.

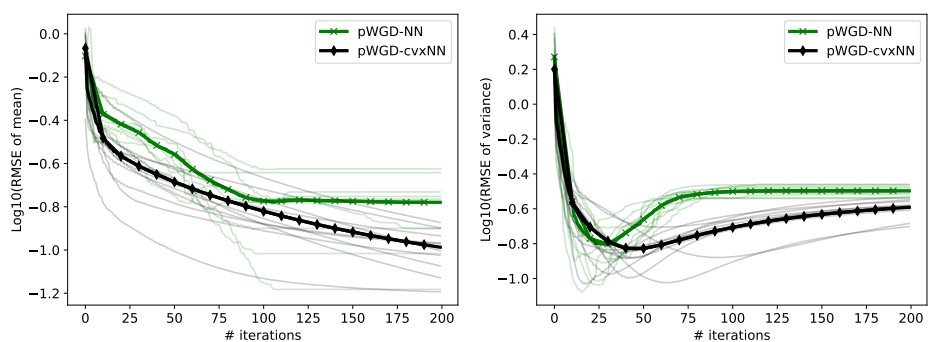

Figure 5: Ten trials and the RMSE of the sample mean (top) and sample variance (bottom) by pWGD-NN and pWGD-cvxNN at different iterations. Linear inference problem.

 ## C  Choice of the regularization parameter

 As the constraints in the relaxed dual problem (16) depends on the regularization parameter $\tilde{\beta}$, it is
 possible that for small $\tilde{\beta}$, the relaxed dual problem (16) is infeasible. Consider the following SDP

$$\min \tilde{\beta}, \text{ s.t. } \tilde{A}_j(\Lambda) + \tilde{B}_j + \sum_{n=0}^{N} r_n^{(j,-)} H_n^{(j)} + \tilde{\beta} e_{d+1} e_{d+1}^T \succeq 0,$$

$$-\tilde{A}_j(\Lambda) - \tilde{B}_j + \sum_{n=0}^{N} r_n^{(j,+)} H_n^{(j)} + \tilde{\beta} e_{d+1} e_{d+1}^T \succeq 0, \tag{21}$$

$$r^{(j,-)} \geq 0, r^{(j,+)} \geq 0, j \in [p].$$

 Here the variables are $\tilde{\beta}$, $\Lambda$ and $\{r^{(j,+)}, r^{(j,-)}\}_{j=1}^p$. Let $\tilde{\beta}_1$ be the optimal value of the above problem.
 Then, only for $\tilde{\beta} \geq \tilde{\beta}_1$, there exists $\Lambda \in \mathbb{R}^{N \times d}$ satisfying the constraints in (16). In other words, the
 relaxed dual problem (16) is feasible. We also note that $\tilde{\beta}_1$ only depends on the samples $X$ and it
 does not depend on the value of $\nabla \log \pi$ evaluated on $x_1, \ldots, x_N$. On the other hand, consider the
 following SDP

$$\min \tilde{\beta}, \text{ s.t. } \tilde{A}_j(Y) + \tilde{B}_j + \sum_{n=0}^{N} r_n^{(j,-)} H_n^{(j)} + \tilde{\beta} e_{d+1} e_{d+1}^T \succeq 0,$$

$$-\tilde{A}_j(Y) - \tilde{B}_j + \sum_{n=0}^{N} r_n^{(j,+)} H_n^{(j)} + \tilde{\beta} e_{d+1} e_{d+1}^T \succeq 0, \tag{22}$$

$$r^{(j,-)} \geq 0, r^{(j,+)} \geq 0, j \in [p],$$

 where the variables are $\tilde{\beta}$ and $\{r^{(j,+)}, r^{(j,-)}\}_{j=1}^p$. Let $\tilde{\beta}_2$ be the optimal value of the above problem.
 For $\tilde{\beta} \geq \tilde{\beta}_2$, as $\mathbf{Y}$ is feasible for the constraints in (16), the optimal value of the relaxed dual problem
 (16) is 0. In short, only when $\tilde{\beta} \in [\tilde{\beta}_1, \tilde{\beta}_2]$, the variational problem (16) is non-trivial. To ensure
 that solving the relaxed dual problem (16) gives a good approximation of the Wasserstein gradient
 direction, we shall avoid choosing $\tilde{\beta}$ either too small or too large.

 ## D  Proofs

 ### D.1  Proof of Proposition 1

 PROOF  We first note that

$$\frac{1}{2} \int \|\nabla \Phi - \nabla \log \rho + \nabla \log \pi\|_2^2 \rho dx$$

$$= \frac{1}{2} \int \|\nabla \Phi\|_2^2 \rho dx + \int \langle \nabla \log \pi - \nabla \log \rho, \nabla \Phi \rangle \rho dx \tag{23}$$

$$+ \frac{1}{2} \int \|\nabla \log \rho - \nabla \log \pi\|_2^2 \rho dx.$$

 We notice that the term $\frac{1}{2} \int \|\nabla \log \rho - \nabla \log \pi\|_2^2 \rho dx$ does not depend on $\Phi$. Utilizing the integration
 by parts, we can compute that

$$\int \langle \nabla \log \rho, \nabla \Phi \rangle \rho dx = \int \left\langle \frac{\nabla \rho}{\rho}, \nabla \Phi \right\rangle \rho dx$$

$$= \int \langle \nabla \rho, \nabla \Phi \rangle dx \tag{24}$$

$$= -\int \Delta \Phi \rho dx.$$

 Therefore, the variational problem (4) is equivalent to

$$\inf_{\Phi \in C^\infty(\mathbb{R}^d)} \frac{1}{2} \int \|\nabla \Phi\|_2^2 \rho dx + \int \langle \nabla \log \pi, \nabla \Phi \rangle \rho dx + \int \Delta \Phi \rho dx. \tag{25}$$

 By restricting the domain $C^\infty(\mathbb{R}^d)$ to $\mathcal{H}$, we complete the proof.

## D.2 Proof of Proposition 2

460 PROOF Suppose that $\hat{w}_i = \beta_i^{-1} w_i$ and $\hat{\alpha}_i = \beta_i^2 \alpha_i$, where $\beta_i > 0$ is a scale parameter for $i \in [m]$.
461 Let $\boldsymbol{\theta}' = \{(\hat{w}_i, \hat{\alpha}_i)\}_{i=1}^m$. We note that

$$\hat{\alpha}_i \hat{w}_i \psi'(\hat{w}_i^T x_n) = \beta_i \alpha_i w_i \psi'\left(\beta_i^{-1} w_i^T x_n\right) = \alpha_i w_i \psi'(w_i^T x_n), \tag{26}$$

462 and

$$\hat{\alpha}_i \|\hat{w}_i\|_2^2 \psi''(\hat{w}_i^T x_n) = \alpha_i \|w_i\|_2^2 \psi''(\hat{w}_i^T x_n) = \alpha_i \|w_i\|_2^2 \psi''(w_i^T x_n). \tag{27}$$

463 This implies that $\Phi_{\boldsymbol{\theta}}(x) = \Phi_{\boldsymbol{\theta}'}(x)$ and $\nabla \cdot \Phi_{\boldsymbol{\theta}}(x) = \nabla \cdot \Phi_{\boldsymbol{\theta}'}(x)$. For the regularization term $R(\boldsymbol{\theta})$,
464 we note that

$$\begin{aligned}
\|\hat{w}_i\|_2^3 + \|\hat{\alpha}_i\|_2^3 &= \beta_i^6 |\alpha_i|^3 + \beta_i^{-3} \|w_i\|_2^3 \\
&= \beta_i^6 |\alpha_i|^3 + \frac{1}{2}\beta_i^{-3}\|w_i\|_2^3 + \frac{1}{2}\beta_i^{-3}\|w_i\|_2^3 \\
&= 3 \cdot 2^{-2/3} \|w_i\|_2^2 |\alpha_i|.
\end{aligned} \tag{28}$$

465 The optimal scaling parameter is given by $\alpha_i = 2^{-1/9} \frac{\|w_i\|_2^{1/3}}{|\alpha_i|_1^{1/3}}$. As the scaling operation does not
466 change $\|w_i\|_2^2 |\alpha_i|$, we can simply let $\|w_i\|_2 = 1$. Thus, the regularization term $\frac{\beta}{2}R(\boldsymbol{\theta})$ becomes
467 $\frac{\tilde{\beta}}{N} \sum_{i=1}^m \|u_i\|_1$. This completes the proof.

## D.3 Proof of Proposition 3

469 PROOF Consider the Lagrangian function

$$\begin{aligned}
L(Z, W, \alpha, \Lambda) =& \frac{1}{2}\|Z\|_F^2 + \sum_{n=1}^N \sum_{i=1}^m \alpha_i \|w_i\|_2^2 \psi''(w_i^T x_n) + \mathrm{tr}(Y^T Z) + \tilde{\beta}\|\alpha\|_1 \\
&+ \sum_{n=1}^N \lambda_n^T \left(z_n - \sum_{i=1}^m \alpha_i w_i \psi'(x_n^T w_i)\right) \\
=& \tilde{\beta}\|\alpha\|_1 + \sum_{i=1}^m \alpha_i \sum_{n=1}^N \left(\|w_i\|_2^2 \psi''(w_i^T x_n) - \lambda_n^T w_i \psi'(x_m^T w_i)\right) \\
&+ \frac{1}{2}\|Z\|_F^2 + \mathrm{tr}((Y+\Lambda)^T Z).
\end{aligned} \tag{29}$$

470 For fixed $W$, the constraints on $Z$ and $\alpha$ are linear and the strong duality holds. Thus, we can
471 exchange the order of $\min_{Z,\alpha}$ and $\max_{\Lambda}$. Thus, we can compute that

$$\begin{aligned}
&\min_{Z,W,\alpha} \max_{\Lambda} L(Z, W, \alpha, \Lambda) \\
=& \min_W \max_\Lambda \min_{\alpha,Z} L(Z, W, \alpha, \Lambda) \\
=& \min_W \max_\Lambda \min_{\alpha,Z} \tilde{\beta}\|\alpha\|_1 + \sum_{i=1}^m \alpha_i \sum_{n=1}^N \left(\|w_i\|_2^2 \psi''(w_i^T x_n) - \lambda_n^T w_i \psi'(x_m^T w_i)\right) + \frac{1}{2}\|Z\|_F^2 + \mathrm{tr}((Y+\Lambda)^T Z) \\
=& \min_W \max_\Lambda -\frac{1}{2}\|\Lambda + Y\|_F^2 + \sum_{i=1}^m \mathbb{I}\left(\max_{w_i:\|w_i\|_2 \le 1} \left|\sum_{n=1}^N \|w_i\|_2^2 \psi''(w_i^T x_n) - y_n^T w_i \psi'(x_n^T w_i)\right| \le \tilde{\beta}\right).
\end{aligned} \tag{30}$$

472 By exchanging the order of $\min$ and $\max$, we can derive the dual problem:

$$\begin{aligned}
&\max_\Lambda \min_W -\frac{1}{2}\|\Lambda + Y\|_F^2 + \sum_{i=1}^m \mathbb{I}\left(\max_{w_i:\|w_i\|_2 \le 1} \left|\sum_{n=1}^N \|w_i\|_2^2 \psi''(w_i^T x_n) - y_n^T w_i \psi'(x_n^T w_i)\right| \le \tilde{\beta}\right) \\
=& \max_\Lambda -\frac{1}{2}\|\Lambda + Y\|_F^2 \text{ s.t. } \max_{w_i:\|w_i\|_2 \le 1} \left|\sum_{n=1}^N \|w_i\|_2^2 \psi''(w_i^T x_n) - y_n^T w_i \psi'(x_n^T w_i)\right| \le \tilde{\beta}, i \in [m] \\
=& \max_\Lambda -\frac{1}{2}\|\Lambda + Y\|_F^2 \text{ s.t. } \max_{w:\|w\|_2 \le 1} \left|\sum_{n=1}^N \|w\|_2^2 \psi''(w^T x_n) - y_n^T w \psi'(x_n^T w)\right| \le \tilde{\beta}, i \in [m]
\end{aligned} \tag{31}$$

473 This completes the proof.

## D.4 Proof of Proposition 4

475 PROOF Based on the hyper-plane arrangements $D_1, \ldots, D_p$, the dual constraint is equivalent to that
476 for all $j \in [p]$,

$$\left| 2 \operatorname{tr}(D_j) \|w\|_2^2 - 2w^T \Lambda^T D_j X w \right| \leq \tilde{\beta} \tag{32}$$

477 holds for all $w \in \mathbb{R}^d$ satisfying $\|w\|_2 \leq 1, (2D_j - I)Xw \geq 0$. This is equivalent to say that for all
478 $j \in [p]$

$$-\tilde{\beta} \geq \min \, 2 \operatorname{tr}(D_j) \|w\|_2^2 - 2w^T \Lambda^T D_j X w, \tag{33}$$
$$\text{s.t. } \|w\|_2 \leq 1, 2(D_j - I)Xw \geq 0,$$
$$\tilde{\beta} \leq \max \, 2 \operatorname{tr}(D_j) \|w\|_2^2 - 2w^T \Lambda^T D_j X w,$$
$$\text{s.t. } \|w\|_2 \leq 1, 2(D_j - I)Xw \geq 0.$$

479 From a convex optimization perspective, the natural idea to interpret the constraint (33) is to transform
480 the minimization problem into a maximization problem. We can rewrite the minimization problem in
481 (33) as a trust region problem with inequality constraints:

$$\min_{w \in \mathbb{R}^d} \, w^T \left( B_j + A_j(\Lambda) \right) w, \tag{34}$$
$$\text{s.t. } \|w\|_2 \leq 1, (2D_j - I)Xw \geq 0.$$

482 As the problem (34) is a convex problem, by taking the dual of (34) w.r.t. $w$, we can transform (34)
483 into a maximization problem. However, as (34) is a trust region problem with inequality constraints,
484 the dual problem of (34) can be very complicated. According to (Jeyakumar & Li, 2014), the optimal
485 value of the problem (34) is bounded by the optimal value of the following SDP

$$\min_{Z \in \mathbb{S}^{d+1}} \, \operatorname{tr}((\tilde{A}_j(\Lambda) + \tilde{B}_j)Z), \tag{35}$$
$$\text{s.t. } \operatorname{tr}(H_n^{(j)}Z) \leq 0, n = 0, \ldots, N,$$
$$Z_{d+1,d+1} = 1, Z \succeq 0.$$

486 from below.

487 **Lemma 1** *The dual problem of SDP* (35) *takes the form*

$$\max -\gamma, \text{ s.t. } S = \tilde{A}_j(\Lambda) + \tilde{B}_j + \sum_{n=0}^{N} r_n H_n^{(j)} + \gamma e_{d+1} e_{d+1}^T, r \geq 0, S \succeq 0, \tag{36}$$

488 *in variables* $r = \begin{bmatrix} r_0 \\ \vdots \\ r_N \end{bmatrix} \in \mathbb{R}^{N+1}$ *and* $\gamma \in \mathbb{R}$.

489 PROOF Consider the Lagrangian

$$L(Z, r, \gamma) = \operatorname{tr}((\tilde{A}_j(y) + \tilde{B}_j)Z) + \sum_{n=0}^{N} r_n \operatorname{tr}(H_n^{(j)}Z) + \gamma(\operatorname{tr}(Z e_{d+1} e_{d+1}^T) - 1), \tag{37}$$

490 where $r \in \mathbb{R}_+^{N+1}$ and $\gamma \in \mathbb{R}$. By minimizing $L(Z, r, \gamma)$ w.r.t. $Z \in \mathbb{S}_+^{d+1}$, we derive the dual problem
491 (36).

492 The constraints on $\Lambda$ in the dual problem (14) include that the optimal value of (35) is bounded from
493 below by $-\tilde{\beta}$. According to Lemma 1, this constraint is equivalent to that there exist $r \in \mathbb{R}^{N+1}$ and
494 $\gamma$ such that

$$-\gamma \geq -\tilde{\beta}, S = \tilde{A}_j(\Lambda) + \tilde{B}_j + \sum_{n=0}^{N} r_n H_n^{(j)} + \gamma e_{d+1} e_{d+1}^T, r \geq 0, S \succeq 0. \tag{38}$$

As $e_{d+1}e_{d+1}^T$ is positive semi-definite, the above condition on $\Lambda$ is also equivalent to that there exist $r \in \mathbb{R}^{N+1}$ such that

$$\tilde{A}_j(\Lambda) + \tilde{B}_j + \sum_{n=0}^{N} r_n H_n^{(j)} + \tilde{\beta} e_{d+1} e_{d+1}^T \succeq 0, r \geq 0. \tag{39}$$

Therefore, the following convex set of $\Lambda$

$$\left\{ \Lambda : \tilde{A}_j(\Lambda) + \tilde{B}_j + \sum_{n=0}^{N} r_n^{(j,-)} H_n^{(j)} + \tilde{\beta} e_{d+1} e_{d+1}^T \succeq 0, \ r^{(j,-)} \geq 0 \right\} \tag{40}$$

is a subset of the set of $\Lambda$ satisfying the dual constraints

$$\left\{ \Lambda : \min_{\|w\|_2 \leq 1, (2D_j - I)w \geq 0} w^T \left( B_j + A_j(\Lambda) \right) w \geq -\tilde{\beta} \right\} \tag{41}$$

On the other hand, the constraint on $\Lambda$

$$\max_{\|w\|_2 \leq 1, (2D_j - I)w \geq 0} w^T \left( B_j + A_j(\Lambda) \right) w \leq \tilde{\beta} \tag{42}$$

is equivalent to

$$\min_{\|w\|_2 \leq 1, (2D_j - I)w \geq 0} -w^T \left( B_j + A_j(\Lambda) \right) w \geq -\tilde{\beta}. \tag{43}$$

By applying the previous analysis on the above trust region problem, the following convex set of $\Lambda$

$$\left\{ \Lambda : -\tilde{A}_j(\Lambda) - \tilde{B}_j + \sum_{n=0}^{N} r_n^{(j,+)} H_n^{(j)} + \tilde{\beta} e_{d+1} e_{d+1}^T \succeq 0, \ r^{(j,+)} \geq 0 \right\} \tag{44}$$

is a subset of the set of $\Lambda$ satisfying the dual constraints

$$\left\{ \Lambda : \max_{\|w\|_2 \leq 1, (2D_j - I)w \geq 0} w^T \left( B_j + A_j(\Lambda) \right) w \leq \tilde{\beta} \right\}. \tag{45}$$

Therefore, replacing the dual constraint $\max_{w:\|w\|_2 \leq 1} \left| \sum_{n=1}^{N} \|w\|_2^2 \psi''(w^T x_n) - y_n^T w \psi'(x_n^T w) \right| \leq \tilde{\beta}$ by

$$\tilde{A}_j(\Lambda) + \tilde{B}_j + \sum_{n=0}^{N} r_n^{(j,-)} H_n^{(j)} + \tilde{\beta} e_{d+1} e_{d+1}^T \succeq 0, j \in [p],$$

$$- \tilde{A}_j(\Lambda) - \tilde{B}_j + \sum_{n=0}^{N} r_n^{(j,+)} H_n^{(j)} + \tilde{\beta} e_{d+1} e_{d+1}^T \succeq 0, j \in [p], \tag{46}$$

$$r^{(j,-)} \geq 0, r^{(j,+)} \geq 0, j \in [p].$$

we obtain the relaxed dual problem. As its feasible domain is a subset of the feasible domain of the dual problem, the optimal value of the relaxed dual problem gives a lower bound for the optimal value of the dual problem.

## D.5 Proof of Proposition 5

PROOF Consider the Lagrangian function

$$L(\Lambda, \mathbf{r}, \mathbf{S}) = -\frac{1}{2}\|\Lambda + Y\|_2^2 - \sum_{j=1}^{p} \text{tr}\left( S^{(j,-)} \left( \tilde{A}_j(\Lambda) + \tilde{B}_j + \sum_{n=0}^{N} r_n^{(j,-)} H_n^{(j)} + \frac{\tilde{\beta}}{2} e_{d+1} e_{d+1}^T \right) \right)$$

$$- \sum_{j=1}^{p} \text{tr}\left( S^{(j,+)} \left( -\tilde{A}_j(\Lambda) - \tilde{B}_j + \sum_{n=0}^{N} r_n^{(j,+)} H_n^{(j)} + \frac{\tilde{\beta}}{2} e_{d+1} e_{d+1}^T \right) \right), \tag{47}$$

where we write

$$\mathbf{r} = \left( r^{(1,-)}, \ldots, r^{(p,-)}, r^{(1,+)}, \ldots, r^{(p,+)} \right) \in \left( \mathbb{R}^{N+1} \right)^{2p},$$

$$\mathbf{S} = \left( S^{(1,-)}, \ldots, S^{(p,-)}, S^{(1,+)}, \ldots, S^{(p,+)} \right) \in \left( \mathbb{S}_+^{d+1} \right)^{2p}. \tag{48}$$

Here we write $\mathbb{S}_+^{d+1} = \{S \in \mathbb{S}^{d+1} | S \succeq 0\}$. By maximizing w.r.t. $\Lambda$ and $\mathbf{r}$, we derive the bi-dual problem (17).

 **D.6  Proof of Theorem 1**

Suppose that $(Z, W, \alpha)$ is a feasible solution to (12). Let $D_{j_1}, \ldots, D_{j_k}$ be the enumeration of $\{\mathbf{diag}(\mathbb{I}(Xw_i \geq 0)) | i \in [m]\}$. For $i \in [k]$, we let

$$S^{(j_i,+)} = \sum_{l:\alpha_l \geq 0, \mathbf{diag}(\mathbb{I}(Xw_l \geq 0))=D_{j_i}} \alpha_l \begin{bmatrix} w_l w_l^T & w_l \\ w_l^T & 1 \end{bmatrix}, S^{(j_i,-)} = 0, \tag{49}$$

and

$$S^{(j_i,+)} = 0, S^{(j_i,-)} = - \sum_{l:\alpha_l < 0, \mathbf{diag}(\mathbb{I}(Xw_l \geq 0))=D_{j_i}} \alpha_l \begin{bmatrix} w_l w_l^T & w_l \\ w_l^T & 1 \end{bmatrix}. \tag{50}$$

For $j \notin \{j_1, \ldots, j_k\}$, we simply set $S^{(j,+)} = 0, S^{(j,-)} = 0$. As $\|w_i\|_2 \leq 1$ and $D_{j_i} = \mathbb{I}(Xw_i \geq 0)$, we can verify that $\mathrm{tr}(S^{(j,-)}H_n^{(j)}) \leq 0, \mathrm{tr}(S^{(j,+)}H_n^{(j)}) \leq 0$ are satisfied for $j = j_1, \ldots, j_m$ and $n = 0, 1, \ldots, N$. This is because for $n = 0$, as $H_0^{(j_i)} = \begin{bmatrix} I_d & 0 \\ 0 & -1 \end{bmatrix}$, it follows that

$$\mathrm{tr}(S^{(j_i,+)}H_0^{(j_i)}) = \sum_{l:\alpha_l \geq 0, \mathbf{diag}(\mathbb{I}(Xw_l \geq 0))=D_{j_i}} \alpha_l(\|w_l\|^2 - 1) \leq 0,$$

$$\mathrm{tr}(S^{(j_i,-)}H_0^{(j_i)}) = - \sum_{l:\alpha_l < 0, \mathbf{diag}(\mathbb{I}(Xw_l \geq 0))=D_{j_i}} \alpha_l(\|w_l\|^2 - 1) \leq 0. \tag{51}$$

For $n = 1, \ldots, N$, we have

$$\mathrm{tr}(S^{(j_i,+)}H_0^{(j_i)}) = \sum_{l:\alpha_l \geq 0, \mathbf{diag}(\mathbb{I}(Xw_l \geq 0))=D_{j_i}} 2\alpha_l(1 - 2(D_{j_i})_{nn})x_n^T w_l \leq 0,$$

$$\mathrm{tr}(S^{(j_i,-)}H_0^{(j_i)}) = - \sum_{l:\alpha_l < 0, \mathbf{diag}(\mathbb{I}(Xw_l \geq 0))=D_{j_i}} \alpha_l(1 - 2(D_{j_i})_{nn})x_n^T w_l \leq 0. \tag{52}$$

Based on the above transformation, we can rewrite the bidual problem in the form of the primal problem (13). For $S \in \mathbb{S}^{d+1}$, we note that

$$\begin{aligned} &\mathrm{tr}(S\tilde{A}_j(\Lambda)) \\ &= -\mathrm{tr}((\Lambda^T D_j X + X^T D_j \Lambda)S_{1:d,1:d}) \\ &= -2\mathrm{tr}(\Lambda^T D_j X S_{1:d,1:d}), \end{aligned}$$

where $S_{1:d,1:d}$ denotes the $d \times d$ block of $S$ consisting the first $d$ rows and columns. This implies that $\tilde{A}_j^*(S) = -2D_j X S_{1:d,1:d}$. Hence, we have

$$\tilde{A}_{j_i}(S^{(j_i,+)} - S^{(j_i,-)}) = - \sum_{l:\mathbf{diag}(\mathbb{I}(Xw_l \geq 0)} 2\alpha_l D_{j_i} Xw_l w_l^T = - \sum_{l:\mathbf{diag}(\mathbb{I}(Xw_l \geq 0)} 2\alpha_l(Xw_l)_+ w_l^T.$$

Therefore, we have

$$\sum_{j=1}^p \tilde{A}_j^*(S^{(j,-)} - S^{(j,+)}) = 2 \sum_{i=1}^m \alpha_i(Xw_i)_+ w_i^T.$$

As $n$-th row of $Z$ satisfies that $z_n = 2\sum_{i=1}^m \alpha_i w_i(x_n^T w_i)_+$, this implies that

$$Z = 2 \sum_{i=1}^m \alpha_i(Xw_i)_+ w_i^T = \sum_{j=1}^p \tilde{A}_j^*(S^{(j,-)} - S^{(j,+)}).$$

Hence $(Z, \{(S^{(j,-)}, (S^{(j,-)}\}_{j=1}^p)$ is feasible to the relaxed bi-dual problem (17).

We can also compute that

$$\sum_{j=1}^p \mathrm{tr}(\tilde{B}_j(S^{(j,+)} - S^{(j,-)})) = 2 \sum_{i=1}^m \alpha_i \sum_{n=1}^N \mathbb{I}(x_n^T w_i \geq 0)\|w_i\|_2^2,$$

and
$$\sum_{j=1}^{p} \text{tr}\left((S^{(j,+)} + S^{(j,-)})e_{d+1}e_{d+1}^{T}\right) = \sum_{i=1}^{m} |\alpha_i|.$$

Thus, the primal problem (13) with $(Z, W, \alpha)$ and the relaxed bi-dual problem (17) with $(Z, \{(S^{(j,-)}, (S^{(j,-)}\}_{j=1}^{p})$ have the same objective value.