# OpenReview forum: "Optimal Neural Network Approximations of Wasserstein Gradient Direction via Convex Optimization"
_NeurIPS.cc/2022/Conference — NeurIPS 2022 Submitted_

### Official Review · Reviewer_XtC2 · 2022-06-19

**Rating:** 5
**Confidence:** 4
**Soundness:** 1 poor
**Presentation:** 2 fair
**Contribution:** 1 poor

**Summary:**

Implementability of Wasserstein gradient flows, which find many applications in sampling and numerical PDEs, hinges on the computation of the Wasserstein gradient. This paper proposes to find the best two-layer squared ReLU activation neural network approximation for the Wasserstein gradient direction. It is shown that this approximation can be formulated as a convex program, albeit with too many constraints to be tractable. The program is then solved approximately by randomly sampling a subset of the constraints. Experiments show that the resulting algorithm has reasonable performance.

**Questions:**

As mentioned above, can you provide runtime comparisons for the experiments?

Also, the formulation of the optimization problem in (5) is similar to work on score matching, see, e.g., [H05], which is used extensively in generative modeling. Can you cite this literature in your work and include a comparison and discussion in the main text?

[H05] Aapo Hyvärinen, Estimation of non-normalized statistical models by score matching.


**Limitations:**

No, the authors do not thoroughly discuss the limitations of their approach.

**Strengths And Weaknesses:**

Seeking new implementations of Wasserstein gradient flows is an important practical problem, and in that respect the approach of this paper is novel and interesting. However, it also seems to fall short when evaluated via either theoretical interest or practical utility.

Regarding practical utility: although the experiments are reasonable, they are largely toy experiments. There is no discussion of the runtime of the method, which leads me to believe that the method is not scalable (since each step requires solving a convex program, it seems computationally heavy). In applications, it appears that this approach yields no benefits over simply parameterizing the Wasserstein gradient via neural networks and training the neural network directly, rather than solving a convex program.

Therefore, it seems more appropriate to evaluate the paper on its theoretical merits, but unfortunately there is not much to speak of. One of the main claims of the paper is that one can use principled convex optimization solvers to implement the method, but in the end the convex program cannot be implemented directly and an approximation must be used. Moreover, with the use of a convex program, one hopes that it comes with some theoretical guarantees, but there are none to be found here.

In short, this paper does not have enough substance. Although the idea is promising, in its current stage the submission is premature and needs more work.

I have also found numerous grammatical errors in the writing, so it would benefit from another round of proofreading.

---

> ### Author Response · Authors · 2022-08-02
> **Response to Reviewer XtC2**
>
> We are grateful to the reviewer's careful reading and constructive comments. In the numerical results, we respectfully disagree with the statement that "they are largely toy experiments". The PDE-constraint Bayesian inference problem is widely applied to estimate the permeability in groundwater flow, thermal conductivity in material science, electrical impedance in medical imaging, etc. The Covid-19 Bayesian inference problem is important to estimate the social distancing parameter from the observed hospitalized cases. For the discussion of the runtime, we do comment on the runtime at the beginning of the numerical experiment. We provide a general response for the runtime of our method. We also respectfully disagree with that "this approach yeilds no benefits over simply parameterizing ...". According to the numerical results, our method WGD-cvxNN does achieve significantly better performance than the non-convex neural network based WGD-NN.
>
> In the theoretical part, we present the calculation of the convex dual problem and derive the relaxation of the dual problem as an SDP. The approximation using subsampling is a commonly used technique in solving the convex optimization problem arising from two-layer neural network, see [Implicit convex regularizers of cnn architectures: Convex optimization of two-and three-layer networks in polynomial time, 2020]. Our main contribution is to construct a convex SDP problem whose optimal solution approximates the Wasserstein gradient direction. We note that the global optimum can be obtained by sampling sufficiently many variables or enumerating all the arrangement patterns. This can be done in polynomial time when the dimension $d$ is a constant. We note that this is a novel result, which can lead to further theoretical investigation of Wasserstein gradient direction approximation due to the convex parameterization.  Otherwise, the subsampling procedure results in a upper-bounding approximation to the convex program. The error due to subsampling can be theoretically analyzed, however, this is left for future research.
> On the other hand, simply fitting a neural network to solve the variational problem using non-convex optimization heuristics (e.g., SGD, ADAM) may reach local minima or saddle points, whose properties are untractable to analyze. By solving the convex program, we always obtain a global solution to the relaxed SDP, whose properties (e.g., stability, robustness, generalization) can be further investigated in future research.
>
> We corrected the grammatical errors in the revision. We also cite [H05] and make the comparison in the revision.

---

> > ### Comment · Reviewer_XtC2 · 2022-08-06
> > **Response**
> >
> > Thank you for the response. Although I still have my doubts about the utility of this approach, I will follow the consensus reached by the other reviewers.

---

> > > ### Author Response · Authors · 2022-08-08
> > > **Response**
> > >
> > > Dear Reviewer XtC2,
> > >
> > > We thank you for your decision to follow the concensus of other reviews, with all ratings to accept the paper. We appreciate if you can adjust your rating accordingly.
> > >
> > > We thank you for your valuable time spent reviewing our work, and we really hope to have a further discussion with you to see if our response resolved your concerns.
> > >
> > > We would appreciate it if you could kindly share your thoughts on the key points in our response, and keep the discussion rolling in case you have further comments. Thank you!
> > >
> > > Best wishes,
> > >
> > > Authors

---

> > > > ### Comment · Reviewer_XtC2 · 2022-08-08
> > > > **Response**
> > > >
> > > > I have updated my score accordingly.

---

### Official Review · Reviewer_oupq · 2022-07-01

**Rating:** 5
**Confidence:** 3
**Soundness:** 3 good
**Presentation:** 2 fair
**Contribution:** 3 good

**Summary:**

This paper is to solve the Wasserstein gradient direction, which is the velocity in Wasserstein gradient flow, without training neural networks. The algorithm is based on a forward discretization of the Wasserstein gradient flow, so once the gradient direction is solved, you can calculate the positions of the particles in the next step. To solve the gradient direction, they firstly construct a least-square regression problem, and add a third-order polynomial regularization term. Then they derive the dual problem in several steps. Finally, they obtain a semi-definite programming relaxation problem and show the duality gap between the relaxed dual problem and the regularized dual problem is zero. The paper provides several illustrative examples.

**Questions:**

**Question/suggestion**

I suggest moving some intermediate propositions to the supplementary materials and adding more transitional contexts, for example, between theorem 1 and the paragraph afterward. The authors can also add more explanations to the notation, e.g. what's the difference between $r^{(j,-)}$ and $r^{(j,+)}$? And I suggest presenting the optimization variables right below min or max in all the optimization problems, like in eq 16 and eq 17.

The dimension reduction technique (row 200) is quite important to accelerate the algorithm. I suggest the authors add more details there.

What does p mean in pWGD?

What does the y-axis mean in the social distancing plot in Figure 4? The social distance in meters?

I suggest adding a reference in section 4.2 about the equation of PDE.

In experiments, different examples use different baseline methods to compare, which is confusing. Why do you use so many different baselines?

In Figure 1, how do you calculate the posterior density? Do you use kernel density estimation or you can precisely calculate it?

**Typo:**

In Figure 4 captions, it should be left / right instead of top/bottom.

In row 210 bracket, compared to WGD-NN only

Overall, I think the writing of this paper can be improved. For example, the equations can be presented in a more concise manner. And the computational complexity needs to be more carefully analyzed.

**Limitations:**

discussed in weakness

**Strengths And Weaknesses:**

**Strength:**

The method can obtain the velocity in Wasserstein gradient flow without training a neural network. They transform the primal problem based on a two-layer neural network into a SDP problem, which can enjoy a similar functional class richness. Unlike SGD in training neural networks, this algorithm can find the global minimum.

The paper gives rigorous derivation toward a relaxed SDP dual problem.

**Weakness:**

The primal problem is regularized, which introduces some bias to the gradient direction.

Several limitations: the target function is only for KL divergence. The function class is only two-layer neural networks.

There is not enough discussion on the computational complexity. I'm rather interested in the complexity dependency on the dimension d, neural network width m, and the number of particles.

The main advantage of this method is no need to train the neural network, which can be very time-consuming. However, the method even takes a longer time than training nn, which makes me a bit disappointed. Can you add more explanations about why WGD-cvxNN takes much longer time?

---

> ### Author Response · Authors · 2022-08-02
> **Response to Reviewer oupq**
>
> We are grateful to the reviewer's careful reading and constructive comments.
>
> The regularization in the primal problem is necessary. Otherwise the optimal value of the primal problem can be $-\infty$ as the corresponding dual problem is infeasible, even when the regularization parameter is small.
>
> Most of the Bayesian inference methods focus on the KL divergence as the objective function. We note that the function class of two-layer neural network itself is rich enough to learn complicated functions. However, our method can be extended to deeper architectures by following the recent work on convex three-layer and deeper networks. This is left for future research.
>
> We provided a general response for the computational costs and clarified this point in the revision.
>
> The reason why WGD-cvxNN takes much longer time is because WGD-cvxNN solves for the global optimum of the dual problem, which can takes more time than training a neural network to solve the primal problem.
>
> For the difference between $r^{(j,+)}$ and $r^{(j,-)}$, we note that  $r^{(j,+)}$ and $r^{(j,-)}$ are different optimization variables. We write the optimization variables below the min or max for all optimization problem in the revision.
>
> Regarding the dimension reduction technique, as it is thoroughly discussed in previous works and our main contribution is the convex formulation of the variational problem of approximating the Wasserstein gradient direction, we briefly introduce these techniques. In pWGD, the letter p stands for 'projected'.
>
> The y-axis in the social distancing plot in Figure 4 means the value of the social distance parameter to infer, which takes value in $[0,1]$.
>
> We add the reference in section 4.2 about the equation of PDE.
>
> In Figure 1, the posterior density can be calculated up to the scale of the normalization constant.

---

> > ### Comment · Reviewer_oupq · 2022-08-06
> > **reply after rebuttal**
> >
> > Thank you for your clarifications. Now I feel the main bottleneck is the complexity dependence over dimension and the number of particles, but this method provides a way to find the global minimum of approximation problem. I would like to increase my score.

---

### Official Review · Reviewer_wk3f · 2022-07-11

**Rating:** 6
**Confidence:** 3
**Soundness:** 3 good
**Presentation:** 3 good
**Contribution:** 2 fair

**Summary:**

The paper considers the problem of implementing Wasserstein gradient flow for the objective of sampling from a target distribution i.e. minimizing KL divergence. It is common to implement the Wasserstein gradient flow with a deterministic system of particles. For minimizing KL divergence, the vector-field  that governs the particles involves the interaction term $\nabla \log(p)$ where $p$ is the distribution of the particles. The most challenging step is to approximate $\nabla \log(p)$ in terms of particles, i.e. the direction of the gradient. This has been subject of numerous recent works in the literature. The authors consider the score-function minimization approach, which involves a stochastic optimization problem over space of functions. The novelty of the work is in the particular choice to representation the functions  and the optimization algorithm to approximate the optimal one. In particular, the authors propose to use a 2-layer neural network representation, consider the dual of the regularized version of the problem, and introduce a convex relation that produces an approximate solution to the original problem. The proposed approach is then numerically evaluated on several benchmark sampling examples.

**Questions:**

Questions:
1- Can you be more explicit and provide a discussion on the computational time of the proposed approach and how it scales with problem parameters?
2- Are there any tuning happening for NN and CVX_NN approach?
3- Does it help if you exclude estimating \nabla $\log (\pi)$ from the formulation because it is known?
4- Can you say anything about the convergence of the gradient flow if the optimization problem to compute the gradient direction is solved up to a certain error?
5- Can you be more explicit about the connection between the exact solution obtained from solving the relaxed convex dual problem and the exact solution to the original problem?



**Limitations:**

Yes

**Strengths And Weaknesses:**

Strength:
- The paper is well-written and clear
- Although there are numerous works on the topic, the topic is still interesting and there are open questions
- The idea of using this particular convexification approach in approximating the Wasserstein gradient is new and interesting

Weakness:
- Computationally expensive approach (in its current form)
- lack of theoretical discussion regarding the final proposed approach

---

> ### Author Response · Authors · 2022-08-02
> **Response to Reviewer wk3f**
>
> We are grateful to the reviewer's careful reading and constructive comments.
>
> 1. We provided a general response for the computational cost of our proposed method. We address the reviewer's questions as follows.
>
> 2. For NN and CVX\_NN approach, we tune for different initial regularization parameters $\lambda$ via grid search to obtain the best performance.
>
> 3. The gradient log posterior is important in our formulation. Excluding this term will make $\Lambda=0$ be the optimal solution to the dual problem, which is meaningless.
>
> 4. The convergence of Wasserstein gradient flow in discrete time is an open problem in Bayesian inference. It is worth an extensive study in future research.
>
> 5. As we can map one feasible solution of the primal problem to one feasible solution to the bidual problem without changing the objective value, the optimal value of the primal problem is lower bounded by the optimal value of the bidual problem.

---

> > ### Comment · Reviewer_wk3f · 2022-08-07
> > **Thanks for the response**
> >
> > Thanks for the response and answering my questions.
> >
> > 3. Maybe I was not clear. I meant that in the formulation 5 or 6, you can re-define $\Phi$ as $\Phi - \log(\pi)$. In other words, you do not have to estimate $\nabla \log(p) - \nabla \log(\pi)$, you only need to estimate $\nabla \log(p)$ and take advantage of the fact that $\nabla \log(\pi)$ is available.

---

> > > ### Author Response · Authors · 2022-08-08
> > > **Response to reviewer wk3f**
> > >
> > > Thanks for clarifying the points. Suppose that we re-define $\Phi$ as $\Phi-\log(\pi)$, the induced problem is equivalent to setting $\log(\pi)=0$ in the original formulation. In this case, the objective the dual problem will become $-\|\Lambda\|_F^2$. Thus, the solution maximizing the dual objective can be trivially $\Lambda=0$. Thus, it is better to estimate $\nabla \log(p)-\nabla \log(\pi)$ instead of directly estimating $\nabla \log(p)$.

---

> > > > ### Comment · Reviewer_wk3f · 2022-08-08
> > > > **Confused!**
> > > >
> > > > Maybe I am missing something and misunderstood your method. So, what happens if the target distribution is a uniform distribution?

---

### Official Review · Reviewer_RwmS · 2022-07-17

**Rating:** 6
**Confidence:** 3
**Soundness:** 3 good
**Presentation:** 3 good
**Contribution:** 3 good

**Summary:**

This paper tackles the variational problem of Wasserstein gradient descent (a.k.a Wasserstein gradient flow with the KL discrepancy). The (variational) optimization problem is restricted to 2-layer neural networks with ReLU activation functions. The main contribution of this paper is to propose a convex (SDP) bi-dual problem that approximates the global solution of the original problem.


**Questions:**

# Questions

1. Theorem 1 shows that the bidual problem can still have feasible solutions with an objective equal to the primal problem: does the mapping go both ways ?
2. Can we uncover additional properties about this mapping between the primal and the bidual ? At least at optimality ?
3. A few references about convexified 2-layer network problems are discussed in 38-41, how does the proposed method relate to these works ?
4. The obtained bidual is an SDP, is this equivalent to the idea of infinite width neural networks modeled as probability measures ?
5. The result obtained in Equation (18) is a bit confusing, Since J corresponds to the bidual loss, I'm not sure what it means about the primal one (which is the one of interest), specially if eq (18) can be a strict inequality. How is that interpretable in terms of "approximating" the global minimum ? Can Z* and tilde(Z*) be analytically compared ? Since this justifies the claim "we solve the global minimization problem", I believe it deserves more clarification.
6. The authors briefly mention the computational burden of the proposed method in L232 being 50x slower than training a neural net. This is not surprising (SDPs are hard to solve) however, I believe this limitation is a very important aspect to investigate. Even though the experiments indicate that lower minima are obtained, the important question of quantifying the ratio:  obtained gain / large computational burden should be answered or at least discussed in more detail. Would such a method still work in large dimensions ? deeper neural networks ? large data sizes ?
7. I'm not sure I follow the point of the Covid-19 experiment, no comment is provided to interpret or analyze the obtained results. Moreover, why pSVGD is considered a reference ?

minor:
 when writing integrals, \int f dx does not make sense: either \int f or \int f(x) dx should be used


typos:
 L52.  we also present *a* practical ..
 L79. In *the* above third ..
 Eq (3): alpha_l is not defined


**Limitations:**

see above

**Strengths And Weaknesses:**

# Strengths and weaknesses
1) Strengths:
- An interesting convexification of a non-convex variational problem, an idea that could be applied in a more general setting
- Even though strong duality does not hold, the authors provide a change of variable (primal -> bi-dual) such that the objective value of feasible solutions are equal

2) Weaknesses:
- The core idea could benefit from more intuitive explanations and comparisons with relatex work (see below)
- The experiments lack the important computational discussion

---

> ### Author Response · Authors · 2022-08-02
> **Response to Reviewer RwmS**
>
> We are grateful to the reviewer's careful reading and constructive comments. For the discussion of the computational cost, we provide a general response.
>
> Regarding the mapping between the primal problem and the bi-dual problem, this mapping cannot go both ways. Otherwise, these two problems are equivalent. As we can map one feasible solution of the primal problem to one feasible solution to the bidual problem without changing the objective value, the optimal value of the primal problem is lower bounded by the optimal value of the bidual problem.
>
> For the convex optimization framework for two-layer neural networks discussed in references 38-41, these works mainly focus on the supervised learning problem of two-layer neural networks using convex loss functions (e.g., squared loss, logistic loss). Our work utilizes a similar convex analytic framework to solve the variational problem of approximating the Wasserstein gradient direction, which is different from supervised learning. The convex optimization approach is related to the idea of infinite width neural networks modeled as probability measures. The dual problem itself is equivalent to the convex dual problem when the neural network in the primal problem has infinitely many neurons. However, the convex optimization approach tackles networks of arbitrary width that are able to learn useful representations, while the infinite width limit is quite limited (limited to basically kernel methods).
>
> For the results in Eq. (18), this means that the optimal value of the primal problem is lower bounded by the optimal value of the bidual problem. We cannot analytically compare $Z^*$ and $\tilde Z^*$ because finding the global optimum of the nonconvex training problem is intrinsically hard.
>
> We admit that this method only works for two-layer networks and the computation cost will can be large when the data dimension or the data size is large. However, our method is able to find the globally optimal solution. Moreover, we obtain significant improvements compared to the baselines in the numerical results.
>
> For the Covid-19 experiment, we provided comments on Figure 4 in Section 4.3. We note that WGD-cvxNN produces more consistent results with pSVGD compared to WGD-NN.
>
> We correct the mentioned minor issue and typos in the revision.

---

### Author Response · Authors · 2022-08-02
**Computation cost and runtime of our proposed method**

Our convex programming formulation involves a standard semi-definite program with $2N\hat p$ inequality constraints and $2\hat p$ linear matrix inequality constraints with size $(d+1)\times(d+1)$, which can be solved using interior-point solvers in at most $O((\max(N,d^2)\hat{p})^6)$ time. Here $\hat p$ is the number of subsampled hyperplane arrangements. We add this discussion of computation complexity in the revision. Meanwhile, though the current runtime of WGD-cvxNN can be much slower than WGD-NN, with specialized designed convex optimizer in future research, we expect the runtime of WGD-cvxNN can be significantly reduced. Besides, for problems where calculating the derivative of the log posterior density is the computation bottleneck, WGD-cvxNN can have competitive runtime compared with WGD-NN.

---

### Comment · Area_Chair_m9ny · 2022-08-03
**Discussion period**

Thanks to all reviewers and authors for their work on this submission.

As the discussion period starts, I want to make sure that reviewers have read the author's response.

This can be done either by communicating with authors, or in private conversation within the reviewing team.

---

### Meta-Review · Area_Chair_m9ny · 2022-08-20

**Recommendation:** Reject
**Confidence:** Certain

**Metareview:**

This work proposes a SDP approach to computing Wasserstein gradient direction for 2-layers NNs, without the need of training the underlying NN. To compute the gradient direction, the authors construct a least-square regression problem, and add a polynomial regularization term. Then, they show that the (relaxed) dual is an SDP problem.

_Pros_
- The idea of casting the Wasserstein gradient direction as an SDP is novel, and interesting. It also paves the way to more general formulations
- The obtained optima is *global*

_Cons_
- The exposition is lacking some motivation at some points. I think the authors could have move some technical discussion (for instance after Thm 1) to give a better insight on the motivations. Some previous works is also sometimes not put in context (for instance regarding the dimensionality reduction).
- For a (mostly) theoretical paper, the statements are sometimes not precise enough, e.g., what is an "equivalent" problem: having the same minimum? the same argmin? In Prop 1, what properties are required on the function space? Don't you need hypothesis on $\psi$? etc.
- From a pratical point of view -- but I don't think that practicality is the core aspect of the paper -- the computational cost is totally prohibitive as discussed by all referees.
- It does not seems that there is a strong practical improvements with respect to training directly the NN after the parameterization of the Wasserstein gradient.

I believe the idea of casting the Wasserstein gradient direction as an SDP problem is interesting, but with respect to the ratio of pros/cons above, and the lack of a strong positive opinion on this work, I recommend to reject this submission in its current state.
I encourage the authors to revise the manuscript in the light of the comments by all reviewers and my own comments for a future submission. In particular, the revision should include the discussion with reviewer RwmS which better highlights your work.

**Award:**

No

---

### Decision · Program_Chairs · 2022-09-14

Reject